# Comprehensive Analysis of the *Hsp20* Gene Family in *Canavalia rosea* Indicates Its Roles in the Response to Multiple Abiotic Stresses and Adaptation to Tropical Coral Islands

**DOI:** 10.3390/ijms23126405

**Published:** 2022-06-08

**Authors:** Mei Zhang, Shuguang Jian, Zhengfeng Wang

**Affiliations:** 1Guangdong Provincial Key Laboratory of Applied Botany and South China Agricultural Plant Molecular Analysis and Genetic Improvement, South China Botanical Garden, Chinese Academy of Sciences, Guangzhou 510650, China; 2CAS Engineering Laboratory for Vegetation Ecosystem Restoration on Islands and Coastal Zones, South China Botanical Garden, Chinese Academy of Sciences, Guangzhou 510650, China; jiansg@scbg.ac.cn; 3Southern Marine Science and Engineering Guangdong Laboratory (Guangzhou), Guangzhou 511458, China

**Keywords:** abiotic stress, *Canavalia rosea*, gene expression, heat shock protein 20

## Abstract

Heat shock protein 20 (Hsp20) is a major family of heat shock proteins that mainly function as molecular chaperones and are markedly accumulated in cells when organisms are subjected to environmental stress, particularly heat. *Canavalia rosea* is an extremophile halophyte with good adaptability to environmental high temperature and is widely distributed in coastal areas or islands in tropical and subtropical regions. In this study, we identified a total of 41 *CrHsp20* genes in the *C. rosea* genome. The gene structures, phylogenetic relationships, chromosome locations, and conserved motifs of each *CrHsp20* or encoding protein were analyzed. The promoters of *CrHsp20*s contained a series of predicted *cis*-acting elements, which indicates that the expression of different *CrHsp20* members is regulated precisely. The expression patterns of the *CrHsp20* family were analyzed by RNA sequencing both at the tissue-specific level and under different abiotic stresses, and were further validated by quantitative reverse transcription PCR. The integrated expression profiles of the *CrHsp20*s indicated that most *CrHsp20* genes were greatly upregulated (up to dozens to thousands of times) after 2 h of heat stress. However, some of the heat-upregulated *CrHsp20* genes showed completely different expression patterns in response to salt, alkaline, or high osmotic stresses, which indicates their potential specific function in mediating the response of *C. rosea* to abiotic stresses. In addition, some of *CrHsp20*s were cloned and functionally characterized for their roles in abiotic stress tolerance in yeast. Taken together, these findings provide a foundation for functionally characterizing *Hsp20*s to unravel their possible roles in the adaptation of this species to tropical coral reefs. Our results also contribute to the understanding of the complexity of the response of *CrHsp20* genes to other abiotic stresses and may help in future studies evaluating the functional characteristics of *CrHsp20*s for crop genetic improvement.

## 1. Introduction

Heat stress (HS) caused by global warming and climate change has become an increasingly significant problem for plant growth [1], seriously affecting plant geographical distribution and crop yields [2]. As sessile organisms, plants cannot avoid HS and are inevitably exposed to complex environmental stimuli [3]. Plants have evolved a series of physiological and molecular strategies to accommodate these stresses over long lifespans, and one of the most important aspects for plants to survive environmental adversity is to maintain cellular homeostasis under stress [4]. Heat shock proteins (Hsps) are molecular chaperones that have been proven to mediate various plant stress responses and regulate plant growth and development. Hsps participate in the folding, localization, repairing, and degradation of proteins in all living organisms and play different roles in protein homeostasis under diverse stresses [5].

Based on their molecular weight and sequence homology, Hsps can be classified into five families: Hsp100, Hsp90, Hsp70, Hsp60, and Hsp20 (also called small Hsp, or sHsp) [6,7]. In plants, Hsp20 was proven to be the most prevalent and abundant family induced by HS [7]; it functions as a molecular chaperone without ATP, which differs obviously from other Hsps (such as Hsp70 or Hsp90). Notably, higher-plant sHsps are particularly diverse and numerous [8], and among all the five conserved Hsp families, Hsp20 is the largest family and the best studied with more plant species being sequenced; it is also the most abundant Hsp found in plants [9]. Hsp20s often have molecular sizes ranging from 15 to 42 kDa, with a highly conserved 80 to 100 amino acid sequence near their C-terminal, which can be defined as the α-crystallin domain (ACD). The typical ACD contains two conserved regions (CRs): CR I with β2, β3, β4, and β5; and CR II with β7, β8, β9, and a β6 loop [10]. The highly conserved ACD is flanked by a variable N-terminal domain (NTD) and a short C-terminal extension (CTE). The NTD participates in substrate binding and the ACD is involved in substrate interactions, while the CTE is responsible for homo-oligomerization [11]. In general, plant Hsp20s can function as molecular chaperones by binding to partially folded or denatured proteins to prevent proteins from irreversible aggregation and keep them stable, thereby preventing damage to proteins caused by environmental stresses. Thus, Hsp20s play important roles in various biotic and abiotic stress responses.

In plants, Hsp20s are divided into various subfamilies (CI-CVI, MTI, MTII, ER, CP, and PX) mainly based on their subcellular localization, sequence homology, and function [7]. The CI-CVI subfamilies are located in the cytoplasm or nucleus; MTI and MTII might work in mitochondrial organelles; and ER, CP, and PX might exist in the endoplasmic reticulum, chloroplasts, and peroxisomes, respectively [12]. In general, Hsp families in plants contain far more members than those in animals [6], which can be attributed to the fact that plants are sessile and have adapted over long periods to different stressful environments.

To date, the Hsp20 gene families were reported in numerous plants, such as Arabidopsis (19) [10], rice (23) [13], pepper (35) [14], tomato (42) [15], and potato (48) [16]. With the advance of whole-genome sequencing (WGS) projects, an increasing number of laboratories around the world have successfully released hundreds of plant genomes [17], and there has been increased focus on some non-model plant species, such as economic plants, specialized-habitat plants, and ornamental plants. Interestingly, the gene regulatory networks (GRNs) revealed by whole genome and transcriptome sequencing in some special-habitat plants have inspired significant research interest in recent years, mainly concerning the adaptation mechanisms regarding extreme adversity [18]. Recent achievements in ‘omics’ technologies are greatly facilitating the assessment of gene families in the regulatory adaptations underlying stress tolerance in plants [19]. *Hsp20* families are essential components of plant tolerance under various abiotic stresses, and the *Hsp20* gene family has been identified in some specialized-habitat plants, including switchgrass [20], willow [21], sorghum [22], and *Sorbus pohuashanensis* [23]. In addition, the functions of some specific plant *Hsp20*s have been further assessed using transgenic methods. For example, a cytosolic class I small Hsp from *Rosa chinensis*, *RcHSP17.8*, could enhance the thermotolerance and resistance to salt, drought, and osmotic stresses of transgenic microorganisms and Arabidopsis [24]. Overexpression of a wheat *TaHSP23.9* in Arabidopsis showed improved heat and salt tolerance in the transgenic lines [25]. The heat-induced and chloroplast-localized sHsp gene from creeping bent grass, *AsHSP26.8a*, displayed slightly improved tolerance to drought, but demonstrated reduced tolerance to heat and salinity stresses when overexpressed in bent grass and Arabidopsis plants [26,27]. Overexpression of a pepper *CaHSP16.4* gene enhanced tolerance to heat stress in Arabidopsis [28]. These studies have clearly demonstrated that Hsp20 genes are involved in regulating a diversity of responses to abiotic stresses, especially heat stress.

Various whole-genome sequencing projects have provided valuable data for improving our understanding of plant Hsp20s, including in the important leguminous crop, soybean, for which the *GmHsp20* family has been systematically characterized [29]. However, little information is available for the *Hsp20* family at the whole-genome level in a specialized habitat leguminous plant with excellent adaptation to extreme adversity. *Canavalia rosea* (Sw.) DC. (Fabaceae) is a representative pantropical coastal plant species with sea-drifted seeds and a wide distribution in tropical and subtropical regions [30]. *Canavalia rosea* exhibits better heat tolerance than most other plant species and great growth potential even under nutrient-deficient conditions on tropical coral reefs. In addition, *C. rosea* is an extremophile halophyte, exhibiting excellent high salt/alkaline and drought tolerance [31], and is, therefore, being used as a ‘pioneering species’ for island greening, sand fixation, and ecological restoration in tropical and subtropical coral islands and coastal zones [32]. Notably, the tremendous growth potential under extreme adversity depends on the long-term adaptive evolution of these specialized habitat species, thereby generating specific GRNs and functional genes. It can be seen that in these specialized habitat plants, such as halophytes [33], the functional identification of genes is of great significance for understanding the mechanisms of adaptive evolution.

In the present study, we comprehensively identified and systematically analyzed the evolutionary relationships, gene duplication events, and the *cis*-acting elements of the promoter regions of the *CrHsp20* family based on the availability of whole-genome sequence data for *C. rosea*. We also further analyzed the expression profiles for the *CrHsp20* gene family members in this specialized habitat leguminous species. In addition, several *CrHsp20* genes in different sub-classes were transcriptionally confirmed by quantitative reverse transcription PCR (qRT-PCR) to evaluate the functions of *CrHsp20* genes in growth, development, and stress responses, especially in the adaptive response to tropical reef ecological environments. Additionally, some heat-responsive *CrHsp20* genes were selected to verify their functions in yeast strains using transgenic technology. The results of this study provide baseline information for the role of an important gene family in the adaptation of a leguminous species to heat, high salt/alkaline, and drought stress, and also offers a reference for gene evolution and function in response to abiotic stresses. These findings will also provide a basis for further functional analyses and for the breeding of stress-tolerant leguminous crops.

## 2. Results

### 2.1. Summary of the CrHsp20 Family in C. rosea

To identify the *CrHsp20* genes in the *C. rosea* genome, the Hidden Markov models (HMMs) profile of the HSP20 domain (PF00011) was taken from the protein family database and searched against the whole-genome database of *C. rosea*. After checking their reliability for acrystalline domains (ACDs) by employing SMART, a total of 41 CrHsp20 protein sequences were ultimately found in the proteome of *C. rosea*. The genomic sequence, transcript sequence, CDS, and protein sequences of the *C. rosea* Hsp20s appear in Appendix A.

The characteristics of the CrHsp20 members, including the gene names, chromosomal locations, open reading frame lengths, amino acid numbers, *M*_W_s, and other physicochemical characteristics [including PI, instability index (II), aliphatic index (AI), and grand average of hydropathicity (GRAVY)] are summarized in Table 1. Overall, the protein lengths of the CrHsp20s varied from 134 (CrHsp20-3) to 693 (CrHsp20-20) amino acids. The *M*_W_s ranged from 14.85 (CrHsp20-3) to 78.654 (CrHsp20-20) kDa. The pI of all CrHsp20s ranged from 5 (CrHsp20-2) to 9.71 (CrHsp20-32), with about 58.5% (24 CrHsp20s) being found to be acidic in nature (pI values < 7, Table 1). The II of the CrHsp20s ranged from 28.89 (CrHsp20-29) to 66.5 (CrHsp20-38), and only 21.95% (nine CrHsp20s) proteins might be considered stable (II < 40). The AI varied between 54.93 (CrHsp20-33) and 105.84 (CrHsp20-6), indicating that these CrHsp20s present different lipophilicities. The majority of the CrHsp20s were hydrophilic in nature, with the GRAVY values being negative (<0), while only CrHsp20-6 might be lipophilic (AI being 105.84 and GRAVY being 0.331). The other 40 CrHsp20s possessed GRAVY values ranging from −0.256 (CrHsp20-32) to −1.166 (CrHsp20-33), indicating that this protein family is quite hydrophilic. According to the WOLF PSORT prediction, most of the CrHsp20s are widely distributed in the whole cell, while only two members, CrHsp20-2 and CrHsp20-40, seem to be localized solely in the nucleus.

### 2.2. Phylogenetic Analysis of the CrHsp20 Gene Family

Based on phylogenetic analysis and previous *Arabidopsis* studies [34], we divided the CrHsp20 members into 14 clusters (Figure 1). The sequence information for these AtsHsps is listed in Appendix A. According to the phylogenetic tree, these 41 CrHsp20s were divided into 11 subfamilies, including 7 cytosol Is (CIs), 5 CIIs, 4 CIIIs, 2 CVs, 2 CVIs, 1 CVII, 3 mitochondria Is (MIs), 3 MIIs, 1 peroxisome (Po/Px), 4 for the endoplasmic reticulum (ER), and 3 plastids (Ps), based on phylogeny and subcellular localization. However, six CrHsp20s (CrHsp20-1, CrHsp20-2, CrHsp20-6, CrHsp20-11, CrHsp20-16, and CrHsp20-22) failed to cluster into any subfamily and were thus unclassified (two clusters in Figure 1). Of the 11 subfamilies, 5 (CIs, CIIs, CIIIs, CVs, CVIs) contained most of the CrHsp20s, indicating that cytoplasm could be the primary functional area of the Hsp20 family in *C. rosea*.

### 2.3. Gene Structures and Motifs of CrHsp20s

The gene structures of all 41 *CrHsp20* genes were predicted from the nucleotide sequences of the cDNA and genomic DNA using a gene structure display server (GSDS 2.0). Unexpectedly, even some gene pairs were highly homologous in sequences, such as *CrHsp20-4*/*CrHsp20-9*, *CrHsp20-8*/*CrHsp20-28*, and *CrHsp20-11*/*CrHsp20-20*, and their gene structures were quite different and possessed different numbers of introns (Figure 2A,B). Of the 41 *CrHsp20*s, only 14 (34.15%) were devoid of introns, while 21 (51.22%) contained 1 intron. Only *CrHsp20-35* had two introns. The *CrHsp20-6* gene consisted of 4 introns, while *CrHsp20-2*, *CrHsp20-14*, *CrHsp20-20*, and *CrHsp20-28* possessed 10, 6, 8, and 11 introns, respectively (Figure 2B).

In addition to the genes’ structures, the conserved motifs in protein sequences of family members also indicated the conservatism and variability of this gene family. To investigate the structural features of the CrHsp20 proteins, the conserved motifs were also analyzed using MEME. A total of 10 distinct motifs, named motif 1 to motif 10, were detected. The lengths of these conserved motifs varied from 15 (motif 9) to 29 (motif 1) amino acids. The sequences of these motifs are listed in Figure 2C. The number of the conserved motifs for each Hsp20 protein ranged from 2 to 6, whereas most CrHsp20s had 4 to 5 conserved motifs (Figure 2C).

### 2.4. Chromosomal Location and Gene Duplication Patterns of CrHsp20 Genes

The *CrHsp20*s were unevenly distributed across 10 of the 11 chromosomes in the *C. rosea* genome (Figure 3). Most of the *CrHsp20*s were located on five chromosomes (Chr1, Chr2, Chr6, Chr7, and Chr11). Chr2 and Chr6 contained the largest numbers of CrHsp20 genes (seven genes on each chromosome). Gene clusters could be observed on Chr6. There were no *CrHsp20*s on Chr10, and only one *CrHsp20* was located on Chr5.

To estimate the evolutionary dates and reveal the functional selection pressure between duplicated gene pairs, the Ka and Ks parameters were calculated (Table 2). A Ka/Ks (the ratio of nonsynonymous substitution to synonymous substitution) value < 1 indicates that a gene pair has experienced purifying selection, whereas Ka/Ks > 1 indicates positive selection, and Ka/Ks = 1 indicates neutral selection. All the Ka/Ks ratios for duplicated gene pairs were less than 0.5, suggesting that they have been subjected to strong purifying selection during evolution.

### 2.5. HSE and Other Stress-Related Cis-Acting Elements in CrHsp20 Promoters

To understand the possible roles of the *cis*-regulatory elements of *CrHsp20* genes relating to plant development and stress-specific responses, we analyzed the promoter region (comprising of −1000 bp upstream of the translation start site) of 41 *CrHsp20*s (Appendix A). Based on their predicted functions, the 12 identified *cis*-regulatory elements were classified into 3 groups, including hormone response, stress response, and transcript factor binding sites (Figure 4, Appendix A). The hormone response elements included gibberellic acid (GA) response, methyl jasmonate (MeJA) response, salicylic acid (SA) response, abscisic acid (ABA) response (ABRE), and ethylene response (ERE); the stress response elements included low-temperature responsive (LTR), elicitor responsive (AT-rich sequence), defense and stress responsive (TC-rich sequence), and pathogen-inducible (as-1); the transcript factor (TF) binding sites included MYB (flavonoid biosynthesis binding site, light binding site, and drought inducible), MYC (dehydration or other abiotic stresses responsive), and Hsf-binding heat shock elements (HSEs).

The promoters of *CrHsp20*s contained a large number of hormone- and stress-related responsive *cis*-acting elements (ERE, ABRE, MYC, and MYB). The EREs appeared most frequently in the promoter regions of the *CrHsp20s*, followed by ABREs (Figure 4A). Comparatively, the defense-related *cis*-acting elements, such as the AT-rich sequence, TC-rich sequence, and as-1, did not occur frequently, indicating that the *CrHsp20* family might not be significantly involved in defense responsiveness (Figure 4). Notably, *C. rosea* plants growing on tropical reefs exhibit better thermotolerance than most species, and so heat stress would be normal during their growth cycle, as would high salinity, alkaline, and drought stresses. Here we found that most of the promoter regions of the *CrHsp20*s processed a number of HSEs, ABREs, MYBs, and MYCs (Figure 4B), indicating that the expression of these gene families could be induced by heat or other stresses and that CrHsp20s might play multifarious protective roles by maintaining cellular homeostasis under different challenges.

### 2.6. Spatiotemporal Expression Patterns of CrHsp20 Genes in Response to Specialized Habitats

First, the RNA-Seq datasets were used to analyze the expression patterns of *CrHsp20*s in different organs of *C. rosea* under normal growth conditions with regular watering and fertilization (plants growing in the South China Botanical Garden, SCBG). The RNA-Seq data of the *CrHsp20*s were generated from five plant tissues, including the root, stem, and leaf from seedlings, and the flower buds and young fruit from mature plants. In general, the expression levels of all *CrHsp20*s in the *C. rosea* leaves seemed to be lowest compared with the other four samples, including root, stem, flower, and young fruit tissues (Figure 5A), which also indicate that *CrHsp20*s, being basic heat responsive genes, maintain low constitutive expression levels in the leaf, since the leaf is the most direct sensor of environmental heat. This was in accordance with the growth conditions, as the inland garden (SCBG) did provide a suitable environment for the *C. rosea* seedlings. Owing to this, we later also extracted the RNA-Seq data from mature leaf samples collected from adult *C. rosea* plants growing in the SCBG and on YX Island (a coral island located in the South China Sea, a native habitat for *C. rosea* plants), and the results show that at least 26 *CrHsp20*s were significantly highly expressed in the leaf samples from YX compared with those from the SCBG, while only several (*CrHsp20-1*, *CrHsp20-24*, and *CrHsp20-31*) showed a slightly opposite trend (Figure 5B). These results indicate that most *CrHsp20*s contribute greatly to the environmental adaptation of *C. rosea* plants to tropical coral islands.

### 2.7. Expression Patterns of CrHsp20 Genes under Abiotic Stresses

Considering the complexity of the environmental conditions of XY, as well as the sampling of *C. rosea* tissues from Yongxing Island not being easy, we separately mimicked the main stressors of YX that could obviously affect plant growth and collected different tissues at different time points. These environmental stressors included: high salinity (600 mM NaCl), alkalinity (150 mM NaHCO_3_ pH 8.2), high osmotic stress (300 mM mannitol, mimicking drought/dehydration), and high temperature (45 °C). To analyze the *CrHsp20*s involved in the response of *C. rosea* to these abiotic stresses, we analyzed the RNA-Seq data to assess the expression levels of the 41 *CrHsp20*s in the roots and leaves during the response to salinity, alkalinity, high osmotic stress, and heat (Figure 6).

Under high salinity stress, only some *CrHsp20*s (*CrHsp20-8*, *CrHsp20-25*, *CrHsp20-26*, and *CrHsp20-28*) showed obviously induced expression under a short-duration challenge in the *C. rosea* roots (2 h), while the levels of induction were further enhanced after a long-duration challenge (48 h). In the roots, alkaline stress also affected the expression of the *CrHsp20* family; however, the influence was smaller than that of salt. Conversely, it appeared that high osmotic stress (drought) caused the most significant change in the *C. rosea* roots at both the time points of 2 h and 48 h, which indicates that the *CrHsp20* genes are deeply involved in the response to water deficit in the roots of *C. rosea* (Figure 6A,B). In the leaves, the expression of the *CrHsp20* family was almost not affected by these three challenges at 2 h, but with the increase in stress duration, these three stresses all caused upregulated expression patterns in some *CrHsp20* members at 48 h, and the alkaline and high osmotic stresses seemed to have more influence than high salinity (Figure 6C,D).

Heat shock for 2 h significantly induced the expression of more than half of the *CrHsp20* members in the *C. rosea* seedlings (Figure 7), and the transcripts of the *CrHsp20*s increased by tens of thousands-fold in both the roots and in the leaves, while several *CrHsp20*s showed slightly downregulated expression patterns in response to heat, including *CrHsp20-1*, *CrHsp20-15*, *CrHsp20-18*, *CrHsp20-19*, *CrHsp20-21*, *CrHsp20-22*, *CrHsp20-23*, *CrHsp20-24*, *CrHsp20-32*, *CrHsp20-33*, *CrHsp20-36*, *CrHsp20-38*, and *CrHsp20-41*. These results further indicate that the *CrHsp20* family might be a key responsive factor involved in the thermotolerance of *C. rosea* plants to their native habitats. The FPKM values for the above RNA-Seq assays are listed in Appendix A.

### 2.8. Expression Analyses of CrHsp20 Genes by qRT-PCR

Based on the RNA-Seq results, we selected 14 *CrHsp20*s for qRT-PCR confirmation. These *CrHsp20*s demonstrated obviously regulated transcriptional patterns during four types of abiotic stress challenge in previous transcriptome analysis (Figure 5B, Figure 6 and Figure 7), and their expression levels in the different tissues were further investigated with gene-specific qRT-PCR assay (Figure 8). The qRT-PCR results show that compared with the salt, alkaline, and high osmotic stresses (dozens to hundreds of times), all 14 transcripts accumulated greatly after the heat stress process (hundreds to thousands, even till to tens of thousands of times) (Figure 8). The qRT-PCR verification results were roughly consistent with the RNA-Seq results, implying that the RNA-Seq data are reliable.

Under salt treatment, the transcript abundance peaked at 48 h in the detected *CrHsp20*s, with the exception of *CrHsp20-5*, *CrHsp20-9*, and *CrHsp20-10*, for which it appeared to decrease at 48 h (Figure 8A,B). The alkaline stress drastically induced the expression of all 14 *CrHsp20*s both in the root and in the leaf, which indicates that these *CrHsp20*s are heavily involved in the response to alkaline toxicity (Figure 8C,D). Similarly, water deficit caused by high osmotic stress also induced the expression levels of *CrHsp20*s, and *CrHsp20-8* presented the highest expression levels both in the roots and in the leaves (Figure 8E,F), suggesting that this gene plays important roles in the response to drought stress. We only performed the short-duration heat challenge (2 h) for the *C. rosea* seedlings, since long-duration heat processing (>4 h) can cause obvious wilting. Our qRT-PCR results also demonstrated that almost all these *CrHsp20*s (except *CrHsp20-14*) showed strikingly apparent expression induction in all three detected tissues (root, stem, and leaf), and their induction degree seemed to be the largest among these abiotic stresses (Figure 8G). The variable expression patterns of these *CrHsp20*s suggests a divergence in biological functions during the adaptation of *C. rosea* to different abiotic challenges. Additionally, the immediate expression induction of *CrHsp20*s also matched the characteristics of *Hsp20s* being stress-responsive genes.

### 2.9. Heterologous Expression of CrHsp20 Confers Heat Tolerance, But Not H_2_O_2_, in Yeast

To further analyze the functions of the *CrHsp20*s, we performed eukaryotic expression analysis in yeast. We successfully cloned seven *CrHsp20* cDNAs, namely *CrHsp20-8*, *CrHsp20-12*, *CrHsp20-13*, *CrHsp20-16*, *CrHsp20-25*, *CrHsp20-26*, and *CrHsp20-27*, and produced *CrHsp20* overexpression yeast vectors with pYES2. *Hsp20* genes were reported to play important roles in mediating intracellular homeostasis under HS challenges both in microorganisms and in plants [35,36].

First, we performed the thermotolerance test for yeast strains overexpressing *CrHsp20*s, and the yeast cultures were subjected to 52 °C for heat stress and shifted to 30 °C for recovery, following which they were gradient-diluted and spotted on SDG plates. Our results demonstrated that the cell viability decreased in the H_2_O_2_-sensitive mutant strain *skn7Δ* harboring the empty vector pYES2 after the HS challenge (52 °C for 15 min), while the WT yeast grew normally. *Skn7Δ* has been confirmed as being sensitive to acute heat stress, probably by acting together with yeast heat shock factors [37]. The *skn7Δ* cells expressing seven *CrHsp20*s always survived better than those harboring pYES2 (Figure 9), indicating that CrHsp20s, being molecular chaperones, are thought to protect the cell during exposure to lethal heat stress, although their exact function remains unclear.

Oxidative stress caused by the increased accumulation of reactive oxygen species (ROS), such as H_2_O_2_ and superoxide anion, is the consequence of many cellular metabolic processes and biotic/abiotic stresses. Hsp20s have significant effects on the response to oxidative stress [38]. Here we also assessed the possible antioxidant abilities of seven CrHsp20s with a yeast spot assay. Unexpectedly, the results showed that all seven *CrHsp20*s failed to relieve H_2_O_2_ toxicity even at relatively low H_2_O_2_ concentrations, either in *skn7∆* (Figure 10A) or in *yap1∆* (Figure 10B), which was quite different from other antioxidant genes, such as *C. rosea CrMT*s [39], or other plant *sHsp* genes, such as *Rosa chinensis RcHSP17.8* [24]. These results indicate that *C. rosea CrHsp20*s might not be directly involved in scavenging ROS but may diminish severe harm from stress in other ways.

## 3. Discussion

Environmental stress is a crucial factor that greatly limits plant growth and affects the distribution of species. On tropical islands or coral reefs, the distribution of plant species is usually highly specific due to the harsh and extreme habitats, including high salinity or alkalinity, seasonal drought, infertile soil, and heat. Hsp20s, which are the most abundant Hsps found in plants, are small molecular chaperones that play important roles in plant growth and development, as well as being widely involved in abiotic stress tolerance, especially to heat stress [6,40,41]. Hsp20s can inhibit the irreversible aggregation of denaturing proteins and enhance cellular thermotolerance, thereby maintaining basic cellular activities and improving the adversity resistance of plants when challenged by heat or other stresses [6]. The halophyte *C. rosea*, belonging to the Fabaceae family, is one seashore halophyte plant that possesses multiple stress-resistant features. This plant has attracted increasing attention due to its strong tolerance to barren soil and can be used as a pioneer species for sand fixation, wind resistance, landscape greening, and ecological restoration in coral islands and coastal zones [31]. Even in extreme heat environments, *C. rosea* presents high biomass yield with minimum requirements for water and nutrition inputs, which suggests that *C. rosea* has evolved unique molecular and physiological mechanisms that allow it to tolerate persistent high temperature [32]. Using good-quality genome assemblies of *C. rosea*, we performed a detailed analysis of the *CrHsp20* family in this species and identified their overall expression patterns, with the aim of elucidating their possible roles in the ecological adaptation of *C. rosea* to tropical coral reefs.

We identified a total of 41 *CrHsp20* genes from the *C. rosea* genome, which could be divided into 11 clusters (CI, CII, CIII, CV, CVI, MI, MII, ER, P, PX/Po, and unclassified). Thirty-nine *Hsp20* genes (*Cc_sHsps*) were identified in another leguminous plant, the pigeon pea [42]. Soybean possesses more *GmHsp20* members in its genome (at least 51 member) [29], probably due to it being an ancient tetraploid that underwent two whole-genome duplications [43]. The classification of plant Hsp clusters is mainly dependent on the variable length N-terminal domain (NTD) or the specific targeting signal peptide (SP) before NTD in the N-terminus of some Hsp20 proteins, because the ACD and CTE located at the middle or C-terminus are more conserved [7,44]. For the phylogenetic tree construction, we only referenced 19 *Arabidopsis* AtHsp20 protein sequences that had been clearly identified and classified into different subfamilies [7,34]. Here we can see that compared with soybean [29] or rice [45] Hsp20 families, the CrHsp20 family also lacks the CIV members that are present in these two species, while the MI and MI subfamilies possess more members in *C. rosea* than they do in soybean and rice (Figure 1). This result indicates that gene acquisition and loss events have occurred in the *CrHsp20* family. Furthermore, gene duplication analysis revealed that both segmental and tandem duplications significantly contributed to the expansion of the *CrHsp20*s (Table 2 and Figure 3), resulting in a relatively large family of *C. rosea Hsp20s*.

The gene structures of the *CrHsp20* family also exhibit characteristics associated with being rapid stress-responsive genes. About 34% (14 of the 41 gene candidates) of the *CrHsp20* genes were intronless, and 51% (21) of the 41 *CrHsp20*s contained only one intron. It is important to note that the occurrence of introns in a gene and the quickly expressing of a gene are controversial, since fewer or no introns were found to enhance the immediate activation of gene expression in plants [46]. The *CrHsp20* genes without or with fewer introns might be a very effective strategy for rapidly responsive gene expression in terms of heat or other abiotic stress challenges. Heat stress often occurs quickly, which inevitably leads to the immediate initiation of heat shock and the induction of gene expression [47,48]. Combined with the physicochemical characteristics of CrHsp20 members (Table 1), about 78% of protein members were unstable (II > 40), and the majority of the CrHsp20s were hydrophilic in nature (GRAVY < 0). Our results indicate that most of the CrHsp20s could be classified as unstable proteins and that they showed rapid induction in response to stress in the cellular environment.

*Hsp20* is a diverse, ancient, and important family present in all organisms [6], and Hsp20 proteins, as molecular chaperones, play important roles in deterring or reducing the irreversible aggregation of denatured proteins in an ATP-independent manner under stress, thereby maintaining cellular homeostasis and regulating plant growth and development [7]. In plants, Hsp20s act as the first line of defense against any adverse condition that could endanger the cellular proteome and protect cells against abiotic stresses, thereby representing an attractive target for sustainable genetic improvement for agricultural production [44]. For instance, various rice *Hsp20* genes are differentially regulated at different developmental stages of rice plants, or are upregulated or downregulated by heat stress or other abiotic and biotic stresses [45]. The gene structure analysis of the *CrHsp20* family indicates that the majority of *CrHsp20* members possess the characteristics of being immediate stress-response genes because they were intronless or only contained one intron (Figure 2), which means their transcription and subsequent translation might be rapidly activated under stress, as the formation of mature mRNA of *CrHsp20*s reduces the steps for the elimination of introns, thus shortening the response time. Gene expression also depends to a great extent on the promoter characteristics. Thus, the *cis*-acting elements in promoter regions play significant roles in gene regulation and expression, and the well-regulated promoters or *cis*-acting elements are strongly required for the engineering of transgenic plants to achieve the controlled expression level of target genes [33]. Investigation of *cis*-acting elements of *CrHsp20* promoter regions indicates that several stress-responsive elements, such as ABRE (ABA responsive element), HSE (heat shock elements), MYB (Myb-binding element), and MYC (Myc-binding element) (Figure 4), are abundant in the promoter regions of most *CrHsp20*s, thereby further defining their possible roles as stress-responsive genes.

Furthermore, the expression patterns of the *CrHsp20*s were investigated on the basis of their transcriptome data from various *C. rosea* tissues collected from different growth habitats, including the SCBG (good growing conditions for *C. rosea* seedlings) and YX (native special habitat). Our results indicate that the *CrHsp20* genes are expressed differentially in various tissues and organs, while under normal conditions, the expression levels of the whole *CrHsp20* family seem to be the lowest (Figure 5A). This is in accordance with plant leaves being the direct sensory organs for heat signaling due to the direct suppressing leaf mitochondrial respiration rate and disturbing leaf cellular energy balance [47,48], and the expression of *Hsp20*s can immediately be modulated by high temperature from a low background level. Numerous studies reported the positive role of *Hsp20*s in plant thermotolerance [24,25,26,27,28], and the typical characteristics of the native habitat of *C. rosea* include perennial high temperatures and strong light conditions. Our RNA-Seq data for the mature leaf tissues collected from YX and the SCBG further support the idea; that is, the majority of *CrHsp20* genes were upregulated in the YX leaf sample (Figure 5B). Given the intensity of sun light and environmental heat, as well as the complicated high salt/alkaline and water shortage on Yongxing Island (YX), this result further implies that *CrHsp20*s might act in abiotic stress responses in the native specialized habitat (YX) of *C. rosea*.

In addition to high temperature, *C. rosea* plants may also need to overcome the high salinity/alkalinity and persistent seasonal drought of sandy soil. As *Hsp20* genes are major stress biomarkers in plants [7,44], a close relationship must exist between the expression of *CrHsp20*s and tolerance to multiple stressors. In this study, we analyzed the expression patterns of *CrHsp20*s in *C. rosea* seedlings in response to various abiotic stresses and found that many differentially expressed *CrHsp20*s were significantly upregulated in response to these stresses, especially to heat (Figure 6 and Figure 7). Similar results were also obtained in other wild plant species growing in special habitats. For example, switchgrass (*Panicum virgatum* L.) is a perennial C4 model grass, and its biomass yield is greatly reduced under controlled or high temperatures. The expression of most *PvHsp20*s responds to heat and other abiotic stresses, and the accumulation of PvHsp20s might facilitate the successful adaptation of switchgrass to the hot and dry plateaus of North America [9]. *Salix suchowensis* is a specialized shrub willow species that presents good resistance to multiple stress conditions. The expression of *Ssu-Hsp20*s was induced by different abiotic stresses, especially heat stress [21]. *Ricinus communis* is an important industrial crop that exhibits good tolerance to a diverse range of abiotic stresses. The *RcsHSP* family showed differentially regulated expression patterns in response to various abiotic stresses, and the overexpression of some *RcsHSP* genes in *A. thaliana* could enhance transgenic seed germination under a variety of abiotic stresses [49]. African bermudagrass (*Cynodon transvaalensis*) is a perennial warm-season turfgrass species that is primarily endemic to some damp and uncultivated areas in South Africa and other regions. The expression patterns of *CtHSP20* genes were differentially regulated by multiple abiotic stresses [50]. In our research, the expression patterns of *CrHsp20*s were investigated in detail in *C. rosea* seedlings subjected to various multiple environmental factors associated with its native habitat, such as heat, high salinity, alkaline, and drought (water deficit). The results from both the RNA-Seq and qRT-PCR assays indicate that, compared with other abiotic stresses, heat stress affects the transcriptional levels of *CrHsp20*s to the greatest extent without exception (Figure 6, Figure 7 and Figure 8). Considering that numerous reports have confirmed the positive correlation between the high expression of *Hsp20*s and good thermotolerance in plants, here we further speculate that the *CrHsp20* family might participate in the regulation of the ecological adaptation of *C. rosea* regarding the multiple adversity on tropical coral islands, especially for extreme heat.

The overexpression of different *Hsp20*s in plants or microorganisms has been demonstrated by multiple studies to elevate stress tolerance [7]. Here we also amplified several *CrHsp20*s from the total cDNAs of the *C. rosea* plants and performed heterologous overexpression with a yeast system—a simple and rapid method for the exploration of the putative functions of genes in vivo. Without exception, all the seven tested *CrHsp20*s enhanced heat tolerance when compared with the control empty vector pYES2 (Figure 9). This is consistent with other plant *Hsp20* genes. For instance, the carrot *Hsp17.7* gene could increase the growth rates and protein solubility in transgenic yeast under heat, cold, acid, and osmotic stress conditions [51]. Three *Camellia sinensis* small Hsp genes, *CsHSP17.7*, *CsHSP18.1*, and *CsHSP21.8*, also increased the tolerance of *Pichia pastori*s to heat and cold through heterologous expression in yeast [52]. Similarly, the overexpression of an *Hsp* (*ThHSP18.3*) from *Tamarix hispida* also conferred salt, alkaline, high osmotic stress, heavy metal, and heat or cold stress tolerance in yeast [53]. These CrHsp20s may act as molecular chaperones to protect yeast against heat stress damage by stabilizing proteins and membranes from degeneration, thereby maintaining regular yeast growth. Although some reports indicated that plant Hsp20s may also reduce the toxic effects of abiotic stresses by scavenging cellular ROS indirectly [24,38,47], and *RcHSP17.8* in rose could enhance yeast H_2_O_2_ tolerance after overexpression, in our study we found that none of these seven CrHsp20s could elevate the H_2_O_2_ tolerance in either the *skn7Δ* or *yap1Δ* yeast mutant strains (Figure 10). This is quite different from other antioxidant molecular chaperones, such as late embryogenesis abundant (LEA) and abscisic acid-, stress-, and ripening-induced (ASR) proteins [54], or metallothioneins (MTs) [39]. Yap1 or Skn7 are two major transcription factors responsive to ROS in yeast, and functional deletion for either of these two regulators in yeast would cause hypersensitivity to H_2_O_2_ [55]. Such differences could be partially interpreted by the fact that H_2_O_2_ toxicity can be directly relaxed by some reductive protein molecules, such as MTs and LEAs/ASRs, while Hsp20s, being molecular chaperones, are mainly in charge of preventing the target proteins or structures from stress-induced aggregation or denaturation, thereby maintaining or recovering cellular homeostasis. On the other hand, although plant Hsp20 involvement was proven in the ROS pathway [47,48], the Hsp20s participate more in the feedback regulation for ROS scavenging, and Hsp protection roles alone cannot cope with excessive ROS production under extreme stress challenges [48]. That is, the ROS scavenging or ROS toxicity- alleviating might be facilitated by accumulating Hsp20 proteins to some extent, while the complete detoxification of ROS has to rely more heavily on the regular-acting cellular antioxidant system, including reductive small molecules (e.g., MTs, LEAs/ASRs) and the antioxidant enzyme system (e.g., superoxide dismutase, peroxidase, catalase). Further work is clearly needed to determine how CrHsp20s act in response to cellular oxidative stress caused by extraneous interference, and more related studies on the effects of such abiotic stresses as high salinity/alkaline, drought, or heavy metals are necessary.

## 4. Materials and Methods

### 4.1. Plant Materials and Stress Treatments

*Canavalia rosea* adult plants growing in the South China Botanical Garden (SCBG, 23°18′76′′ N, 113°37′02′′ E, Guangzhou city, China) had been transplanted gradually from Hainan province, China since 2012, and *C. rosea* adult plants growing in Yongxing Island (YX, 16°83′93′′ N, 112°34′00′′ E) were also used in this study. The *C. rosea* seedlings were cultivated in the SCBG with seeds gathered from the coastal areas of Hainan province. In brief, the *C. rosea* seeds were germinated in pots filled with soil/vermiculite mixture and grown to seedlings for 30 days in a growth chamber or until they were flowering and bearing fruits outdoors in the SCBG. They received regular water and fertilizer.

To perform the RNA-Seq analysis, the *C. rosea* roots, stems, leaves, flowers, and fruits were gathered from seedlings or adult plants growing in the SCBG to obtain the tissue-specific transcriptional patterns of the identified *CrHsp20*s. Adult leaves were then gathered from *C. rosea* plants growing in both YX and the SCBG to investigate the involvement of the *CrHsp20*s in adaptation to different habitats. For abiotic stress-related RNA-Seq analysis, the 30-day-old *C. rosea* seedlings were removed from the pots and carefully cleaned, following which they were transferred into 600 mM NaCl solution (1/2 Hoagland’s) for high-salinity stress. For salt–alkaline stress, the cleaned *C. rosea* seedlings were soaked in a 150 mM NaHCO_3_ (pH 8.2) solution (1/2 Hoagland’s). For high osmotic stress treatment, the seedlings were soaked in a 300 mM mannitol solution (1/2 Hoagland’s). For heat stress (HS), the seedlings were soaked in 45 °C pre-warmed 1/2 Hoagland’s solution with their roots submerged and placed in a 45 °C-thermostatic light incubator for 2 h. The *C. rosea* seedlings placed in 1/2 Hoagland’s solution for different time points (0, 2 h, 48 h) were used as the control. At certain time points (0, 2 h, 48 h), the *C. rosea* tissues were collected and immediately frozen in liquid nitrogen, following which the samples were stored at −80 °C for subsequent RNA-Seq analysis. Three independent biological replicates were used.

### 4.2. Genome-Wide Identification of Hsp20 Proteins and Chromosomal Location of CrHsp20s

The assembled genome data of *C. rosea* (GenBank accession no.: JACXSB000000000, which will be released on 16 September 2024) were annotated with InterPro and Pfam for gene family identification with conserved domains and motifs (e < 1 × 10^−5^). Subsequently, Pfam ID (Hsp20 domain, PF00011) was used to search for CrHsp20 family members (Pfam 34.0; http://pfam.xfam.org/, accessed on 21 March 2022). Finally, candidate *CrHsp20*s were named based on their gene locus referred to in the *C. rosea* genome annotation. The chromosomal position of each *CrHsp20* gene was acquired and mapped from MG2C (MapGene2Chrom, http://mg2c.iask.in/mg2c_v2.0/, accessed on 21 March 2022) according to the whole-genome sequencing information of *C. rosea*.

The coding sequences, genomic sequences, promoter sequences (ATG upstream 1000 bp), and amino acid sequences of the *CrHsp20* genes were downloaded from the *C. rosea* genome database. All the sequences are summarized in Appendix A. The theoretical isoelectric point (pI) and molecular weight (*M*_W_) of each Hsp20 protein were estimated using the pI/*M*_W_ tool from ExPASy (v.3.0; http://web.expasy.org/protparam/, accessed on 21 March 2022), and the 3D structure of CrHsp20 proteins was predicted with the Protein Fold Recognition Server tool (PHYRE^2^, http://www.sbg.bio.ic.ac.uk/phyre2/html/page.cgi?id=index, accessed on 21 March 2022). For the subcellular localization prediction, the online program WoLF_PSORT (https://wolfpsort.hgc.jp/, accessed on 21 March 2022) was used.

The intron–exon structures of the *CrHsp20*s were visualized using the Gene Structure Display Server (GSDS 2.0; http://gsds.cbi.pku.edu.cn, accessed on 21 March 2022). The conserved motifs of the Hsp20 proteins were identified using Multiple Em for Motif Elicitation (MEME) software (http://meme-suite.org/tools/meme, accessed on 21 March 2022), with the parameters of optimum motif width range from 6 to 50 amino acid residues and a maximum of 10 motifs.

### 4.3. Phylogenetic Analysis, Gene Duplication, and Non-Synonymous (Ka) and Synonymous (Ks) Calculation of C. rosea Hsp20 Genes

The amino acid sequences of the Hsp20s derived from *Arabidopsis* [34] and the identified CrHsp20s in this study were used for phylogenetic analysis. A neighbor-joining phylogenetic tree was created based on multiple protein sequence alignments of these Hsp protein sequences with 1000 bootstrap replicates in MEGA (version 6) software (https://www.megasoftware.net/, accessed on 21 March 2022). The gene segmental and tandem duplications were identified with Multiple Collinearity Scan toolkit (MCScanX) software [56], and tandem duplications were also checked manually according to their gene loci. The number of synonymous substitutions per synonymous site (Ka), the number of non-synonymous substitutions per nonsynonymous site (Ks), and the *P*-value from a Fisher’s exact test of neutrality were calculated using the Nei–Gojobori model with 1000 bootstrap replicates [57]. The ratio of Ka/Ks being greater than, equal to, or less than 1, indicates positive, neutral, and purifying selection, respectively. Syntenic relationships and gene duplications of the *CrHsp20*s were evaluated using Circos v. 0.63 (http://circos.ca/, accessed on 21 March 2022).

### 4.4. Analysis of Cis-Acting Elements in CrHsp20 Gene Promoters

The *cis*-acting elements in the 1000 bp upstream sequences (listed in Appendix A) of the coding region of the *C. rosea Hsp20* genes were retrieved from the genome, and the types, numbers, and functions of these elements were analyzed using PlantCARE software (http://bioinformatics.psb.ugent.be/webtools/plantcare/html/, accessed on 21 March 2022). The heat-stress elements (HSEs), including 12 putative sequences that have been confirmed in *Arabidopsis*, rice, maize, and soybean, were also identified manually according to a previous report [58]. These possible conserved HSE sequences included 5′-CTAGAA-3′, 5′-TTCAGA-3′, 5′-TCTAGA-3′, 5′-TCCAGA-3′, 5′-TCTGGA-3′, 5′-CGCTTC-3′, 5′-GAAGCG-3′, 5′-GGTTCG-3′, 5′-CGAACC-3′, 5′-TTCCAG-3′, 5′-CTGGAA-3′, and 5′-TTCNNGAA-3′, and N presents any of the four nucleotide bases. The *cis*-acting elements were summarized with Microsoft Excel v. 2010 software (Microsoft Corp., Albuquerque, NM, USA), and several selected *CrHsp20* promoters were visualized using TBtools.

### 4.5. RNA-Sequencing Data Analysis of CrHsp20 Genes

The RNA-Seq libraries were prepared for all *C. rosea* tissues and were sequenced on an Illumina HiSeq 2000 platform (Oebiotech Company, Shanghai, China). In brief, the transcriptome databases were produced with at least 40 Gb clean reads per sample. Clean reads were mapped to the *C. rosea* reference genome using Tophat v.2.0.10 (http://tophat.cbcb.umd.edu/, accessed on 21 March 2022). The transcriptional levels were calculated against the assembled unigene library, and the heatmaps showing the *CrHsp20* expression profiles were generated using TBtools, principally adopting the fragments per kilobase of transcript per million mapped reads (FPKM) values, and the expression levels (log2 of FPKM values) of these genes were visualized.

### 4.6. Total RNA Extraction and Expression Analyses of CrHsp20 Genes by qRT-PCR

Total RNA was extracted from each *C. rosea* tissue sample using a plant RNA extraction kit (TransGen Biotech, Beijing, China) according to the manufacturer’s instructions. The RNA concentrations were determined using a NanoDrop 1000 (Thermo Fisher Scientific, Waltham, MA, USA) and the quality was assayed on a 1% agarose gel. The cDNAs were obtained by total RNA reverse transcription using AMV reverse transcriptase (CISTRO BIO, Guangzhou, China). To quantify the relative transcript levels of selected *CrHsp20* genes, qRT-PCR was performed with gene-specific primers using the LightCycler480 system (Roche, Basel, Switzerland) and 2 × Ultra SYBR Green qPCR Mix (CISTRO BIO) according to the manufacturers’ instructions. All gene expression data obtained via qRT-PCR were normalized to the expression of *CrEF-1α*. The gene-specific primers used for this analysis are listed in Appendix A.

### 4.7. Cloning of CrHsp20 cDNAs and Functional Identification of CrHsp20s in Yeast

To clone the *CrHsp20* cDNAs, total RNA isolation and cDNA synthesis were performed as described above. The selected full-length *CrHsp20* cDNA was PCR-amplified from *C. rosea* seedling cDNA using the gene-specific primers listed in Appendix A. The yeast expression vector CrHsp20s-pYES2 was constructed with the in-fusion technique (BD In-Fusion PCR cloning Kit, Takara Bio USA, Mountain View, CA, USA) according to the manufacturer’s instructions. In short, the PCR fragments were inserted into the *Bam*HI and *Eco*RI sites of pYES2 and, after sequencing confirmation, these recombinant expression vectors were transformed into different yeast strains to verify the stress tolerance functions. The wild-type (WT) yeast (*Saccharomyces cerevisiae*) strain BY4741 (Y00000, MATa; *ura3Δ0*; *leu2Δ0*; *his3Δ1*; *met15Δ0*) and the two H_2_O_2_-sensitive mutant strains *yap1Δ* (Y00569, BY4741; MATa; *ura3Δ0*; *leu2Δ0*; *his3Δ1*; *met15Δ0*; *YML007w::kanMX4*), and *skn7Δ* (Y02900, BY4741; MATa; *ura3Δ0*; *leu2Δ0*; *his3Δ1*; *met15Δ0*; *YHR206w::kanMX4*) were obtained from Euroscarf (Frankfurt, Germany). The various yeast strains were transformed using the LiOAc/PEG method, and uracil complementation was used for selection. A synthetic defined (SD) yeast medium without uracil (SD-Ura) containing 2% (*w*/*v*) galactose (Gal) was used as the yeast induction medium (SDG-Ura) during the heat-stress tolerance and oxidative stress experiments on solid medium for the localization observations. In brief, the single colonies of yeast transformants were selected and used to inoculate liquid SDG-Ura medium and cultured overnight or longer at 30 °C in a shaker (150–200 rpm), diluted with fresh pre-warmed SDG medium (volume ratio 1:10), and then incubated with continuous shaking for about 30 h at 30 °C to reach an optical density of 1 at OD600 (optical density at 600 nm). The cells were then serially diluted in 10-fold steps, and 2-μL aliquots of each were finally spotted onto SDG medium plates with or without H_2_O_2_ for oxidative stress tolerance. For the thermotolerance test, the yeast cultures were incubated at 52 °C for 15 min (*skn7Δ* and WT) before gradient dilution and spotting on SDG plates. The plates were incubated at 30 °C for 2 to 5 days and then photographed.

## 5. Conclusions

In this study, *C. rosea CrHsp20* genes were systematically and comprehensively analyzed based on genome sequencing data. A total of 41 *CrHsp20*s were identified and their gene structures, gene duplication models, stress-related *cis*-acting elements in promoter regions, and expression patterns in different tissues or under specific abiotic stresses were also systematically explored. Our study provides comprehensive information on the *CrHsp20* gene family and should improve our understanding of the functional divergence of *Hsp20* genes in *C. rosea*. In addition, we discussed the possible roles in which the *CrHsp20* family might be involved in multiple stress responses, especially heat stress. Several *CrHsp20*s were cloned and functionally confirmed in yeast for their possible roles in thermotolerance. The systematic characterization of the *CrHsp20* family not only provides a meaningful overview of this gene family, but also contributes toward illuminating the heat resistant mechanism of this specialized plant and even the adaptability of *C. rosea* to other stress conditions.

## Figures and Tables

**Figure 1 ijms-23-06405-f001:**
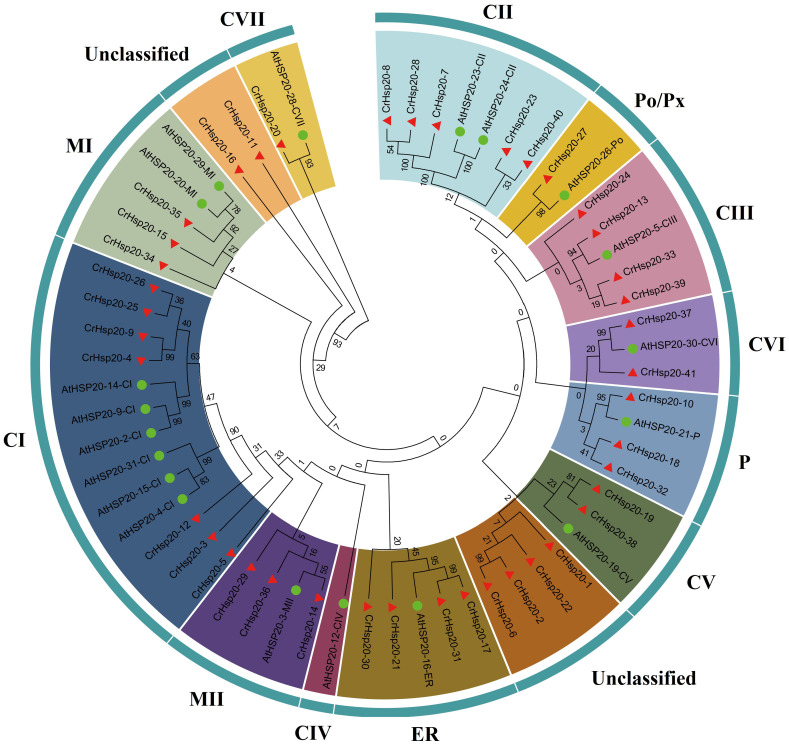
Phylogenetic tree of 41 CrHsp20 proteins from *C. rosea* and nineteen AtHsp20 proteins from Arabidopsis. The Hsp20 subfamilies were distinguished by different colors. The unrooted tree was generated by ClustalW in MEGA6 based on the NJ method; bootstrap was 1000 replicates.

**Figure 2 ijms-23-06405-f002:**
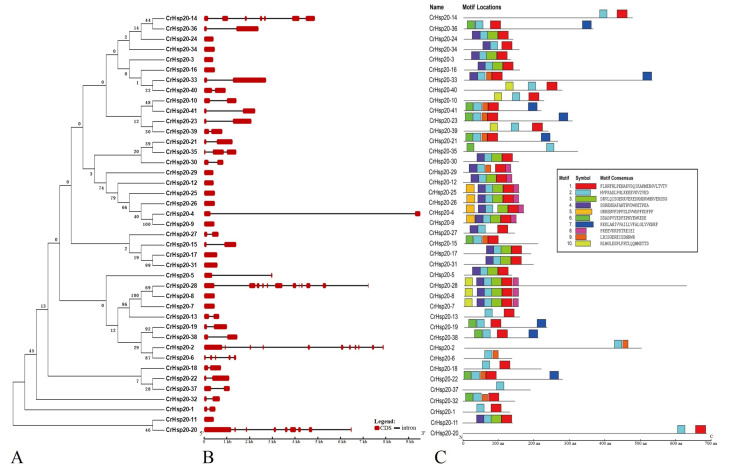
Exon–intron structures of *CrHsp20* genes and conserved motifs in CrHsp20 proteins. (**A**) Phylogenetic tree of 41 CrHsp20s was constructed with MEGA6 (NJ method; bootstrap being 1000 replicates). (**B**) Gene structure of *CrHsp20*s. The lengths and positions of introns and exons are shown with black lines and red round-corner rectangles for CrHsp20 genes ordered by their names. (**C**) Domain analyses of CrHsp20 proteins. Different color boxes represent the different types of motifs. These motifs were identified using Multiple EM for Motif Elicitation (MEME) and boxes with different colors represent different motifs.

**Figure 3 ijms-23-06405-f003:**
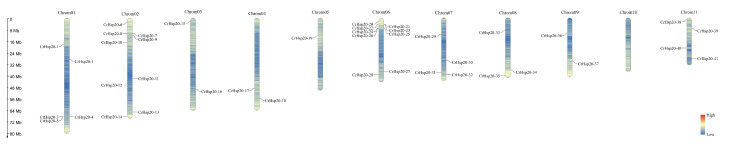
Distribution of *CrHsp20* genes in *C. rosea* chromosomes. Forty-one *CrHsp20* genes were mapped to the 10 chromosomes, whereas chromosome 10 showed no *CrHsp20* location. The chromosome number is listed at the top of each chromosome, and the scale of the chromosome is in millions of bases (Mb).

**Figure 4 ijms-23-06405-f004:**
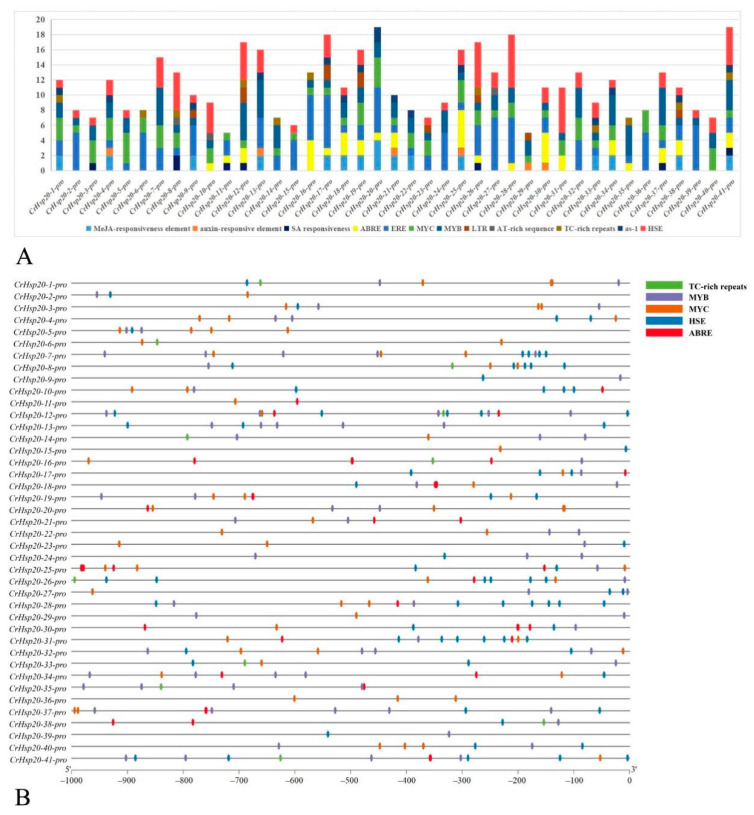
Predicted *cis*-acting elements in *CrHsp20* promoter regions of *C. rosea*. The 1000 bp sequence upstream of 41 *CrHsp20* coding region was analyzed by PlantCARE and manually searched with HSEs. The HSEs are the sum total of 12 sequence motifs mentioned in subsequent “Materials and Methods” part, including all of possible conserved sequences (5′-CTAGAA-3′, 5′-TTCAGA-3′, 5′-TCTAGA-3′, 5′-TCCAGA-3′, 5′-TCTGGA-3′, 5′-CGCTTC-3′, 5′-GAAGCG-3′, 5′-GGTTCG-3′, 5′-CGAACC-3′, 5′-TTCCAG-3′, 5′-CTGGAA-3′, and 5′-TTCNNGAA-3′, N presents any of four nucleotide bases), which have been confirmed in Arabidopsis, soybean, rice, and maize. (**A**) Summaries of the 11 *cis*-acting elements in 41 *CrHsp20* promoter regions. The abundance of different elements on each of the promoters is shown in different colors. (**B**) The location of abiotic stress-responding cis-acting elements (TC-rich repeats, MYB, MYC, HSE, and ABRE) in 41 *CrHsp20* promoter regions.

**Figure 5 ijms-23-06405-f005:**
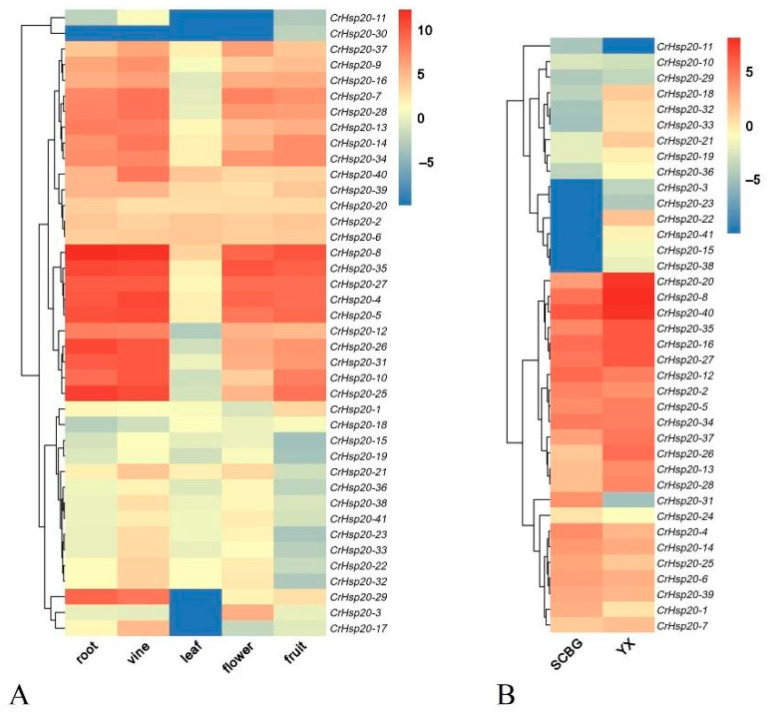
Heatmaps showing (**A**) the expression levels of the 41 *CrHsp20*s in the root, stem, leaf, flower bud, and young fruit of *C. rosea* plant and (**B**) expression differences of the 41 *CrHsp20*s in mature *C. rosea* leaves planting in South China Botanical Garden (SCBG) and in Yongxing Island (YX). The expression level of each gene is shown in FPKM (log2 values). Red denotes high expression levels and blue denotes low expression levels. The heatmaps were constructed by oebiotech cloud (https://cloud.oebiotech.cn/task/detail/heatmap/, accessed on 21 March 2022).

**Figure 6 ijms-23-06405-f006:**
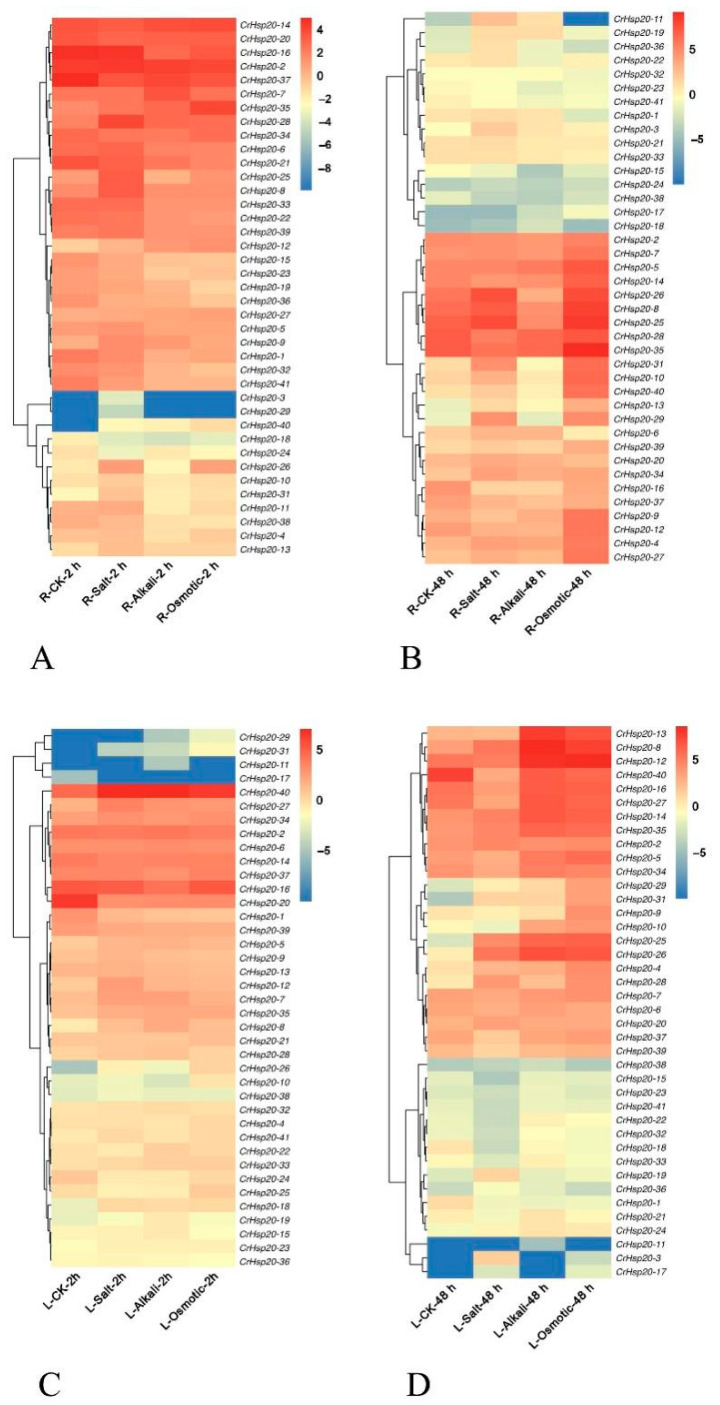
Heatmaps showing the expression patterns of the *CrHsp20*s under abiotic stress treatment (600 mM NaCl, 150 mM NaHCO_3_, pH 8.2, and 300 mM mannitol) in roots (**A**,**B**) and leaves (**C**,**D**) after 2 h challenge (**A**,**C**) or 48 h challenge (**B**,**D**). R: roots; L: leaves; CK: control; Salt: 600 mM NaCl; Alkali: 150 mM NaHCO_3_, pH 8.2; Osmotic: 300 mM mannitol. The expression level of each gene is shown in FPKM (log2 values). Red denotes high expression levels and blue denotes low expression levels. The heatmaps were constructed by oebiotech cloud (https://cloud.oebiotech.cn/task/detail/heatmap/, accessed on 21 March 2022).

**Figure 7 ijms-23-06405-f007:**
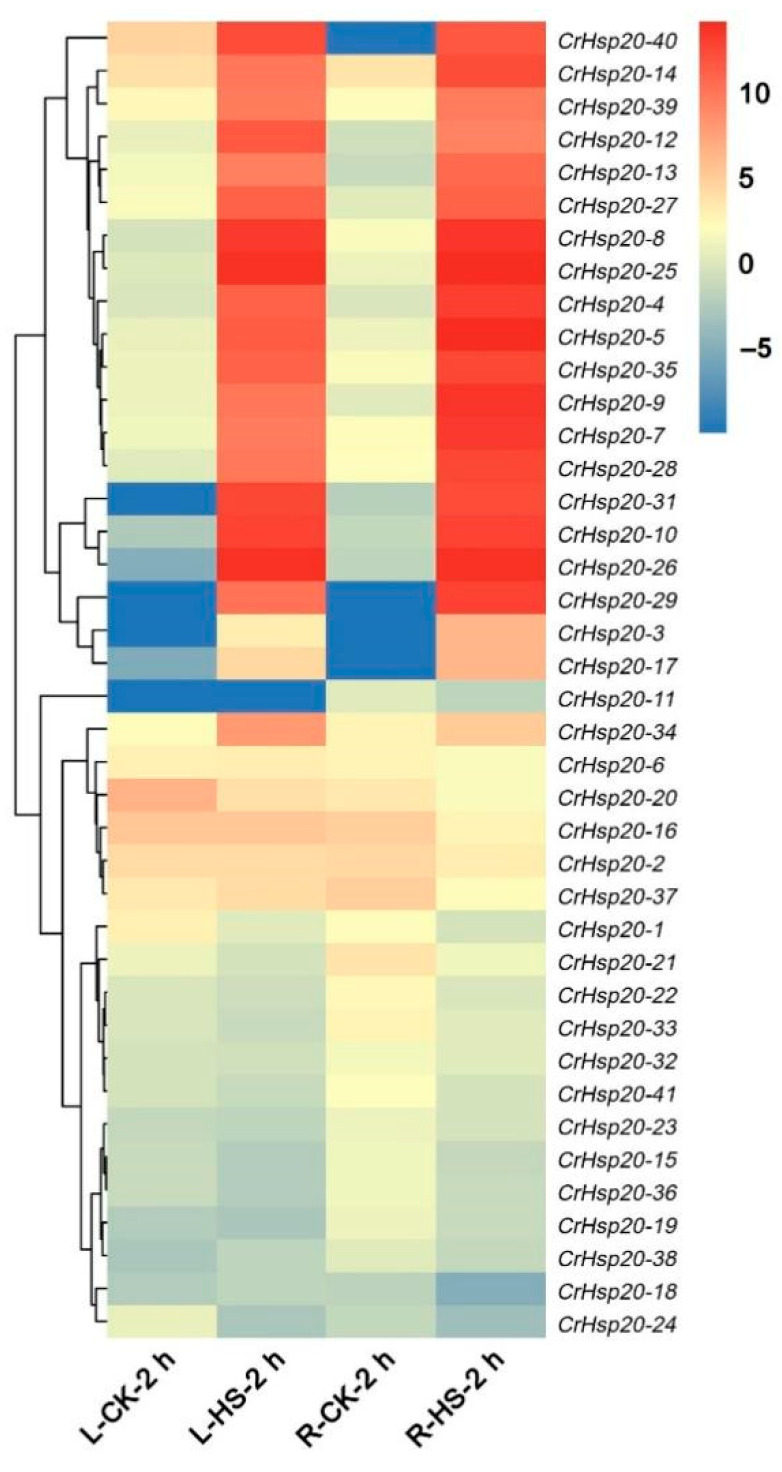
Heatmap showing the expression patterns of the *CrHsp20*s under 2 h heat stress (45 °C) in leaf or root samples. R: roots; L: leaves; CK: control; HS: heat stress (45 °C). The expression level of each gene is shown in FPKM (log2 values). Red denotes high expression levels and blue denotes low expression levels. The heatmaps were constructed by oebiotech cloud (https://cloud.oebiotech.cn/task/detail/heatmap/, accessed on 21 March 2022).

**Figure 8 ijms-23-06405-f008:**
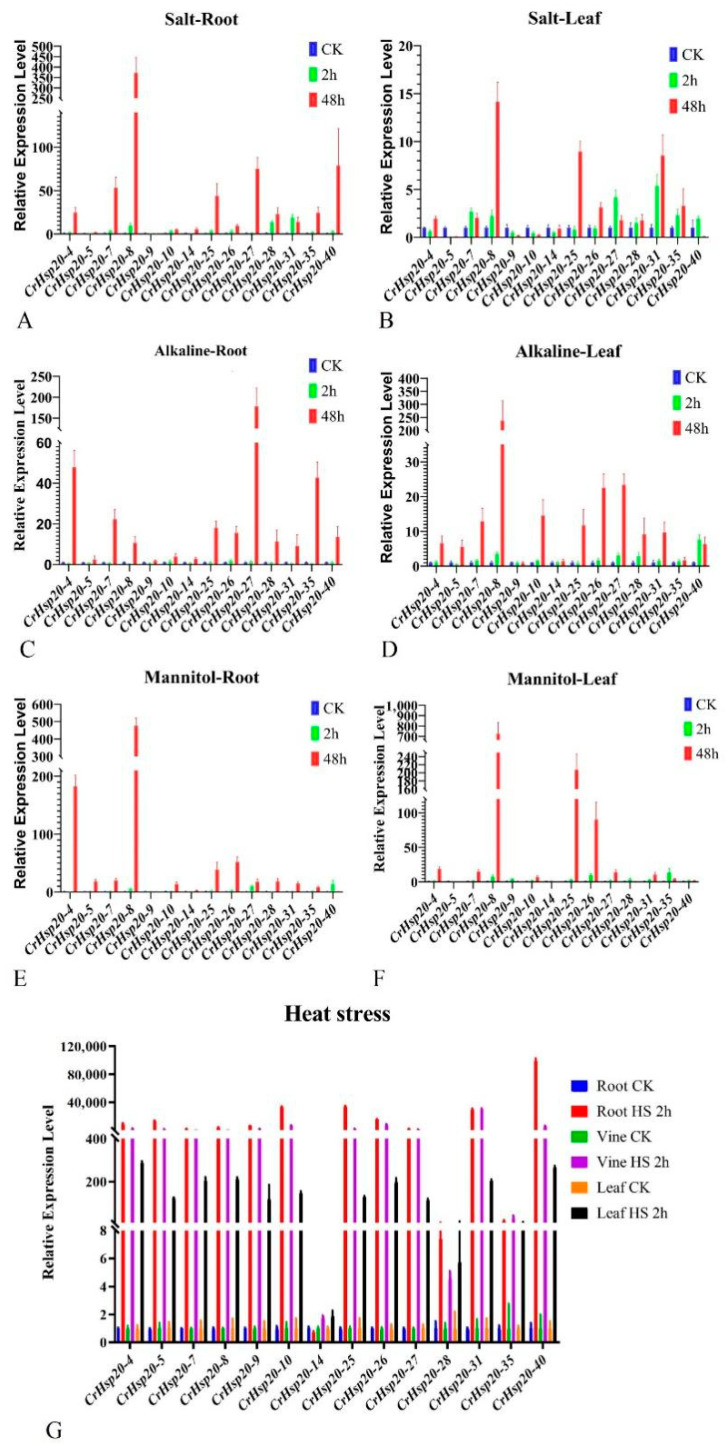
Quantitative RT-PCR detection of the expression levels of the 14 CrHsp20s (CrHsp20-4, CrHsp20-5, CrHsp20-7, CrHsp20-8, CrHsp20-9, CrHsp20-10, CrHsp20-14, CrHsp20-25, CrHsp20-26, CrHsp20-27, CrHsp20-28, CrHsp20-31, CrHsp20-35, and CrHsp20-40) responding to different stresses (600 mM NaCl for salt stress, 150 mM NaHCO_3_ for alkaline stress, 300 mM mannitol for high osmotic stress, or heat stress at 45 °C) in C. rosea seedling plants. (**A**) The expression patterns in roots under salt treatment. (**B**) The expression patterns in leaves under salt treatment. (**C**) The expression patterns in roots under alkaline treatment. (**D**) The expression patterns in leaves under alkaline treatment. (**E**) The expression patterns in roots under high osmotic stress treatment. (**F**) The expression patterns in leaves under high osmotic stress treatment. (**G**) The expression patterns in roots, stems, and leaves under heat stress. Relative expression values were calculated using the 2^−ΔCt^ method with housekeeping gene CrEF-1α as reference gene.

**Figure 9 ijms-23-06405-f009:**
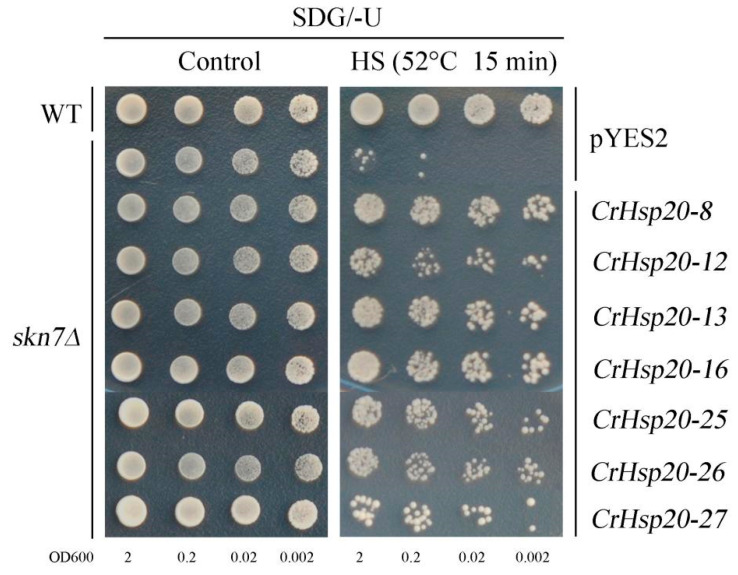
The thermotolerance confirmation of seven CrHsp20s (CrHsp20-8, CrHsp20-12, CrHsp20-13, CrHsp20-16, CrHsp20-25, CrHsp20-26, and CrHsp20-27) using heteroexpression in yeast. The yeast cultures were adjusted to OD600 = 2 firstly, then were incubated at 52 °C for 15 min and moved to 30 °C environment before being spotted on SDG/−U medium (without any chemical stress factors) for thermotolerance confirmation. The corresponding yeast spots growing on SDG/−U plates without heat stress were used as the control. The plates were incubated for 2–5 days at 30 °C.

**Figure 10 ijms-23-06405-f010:**
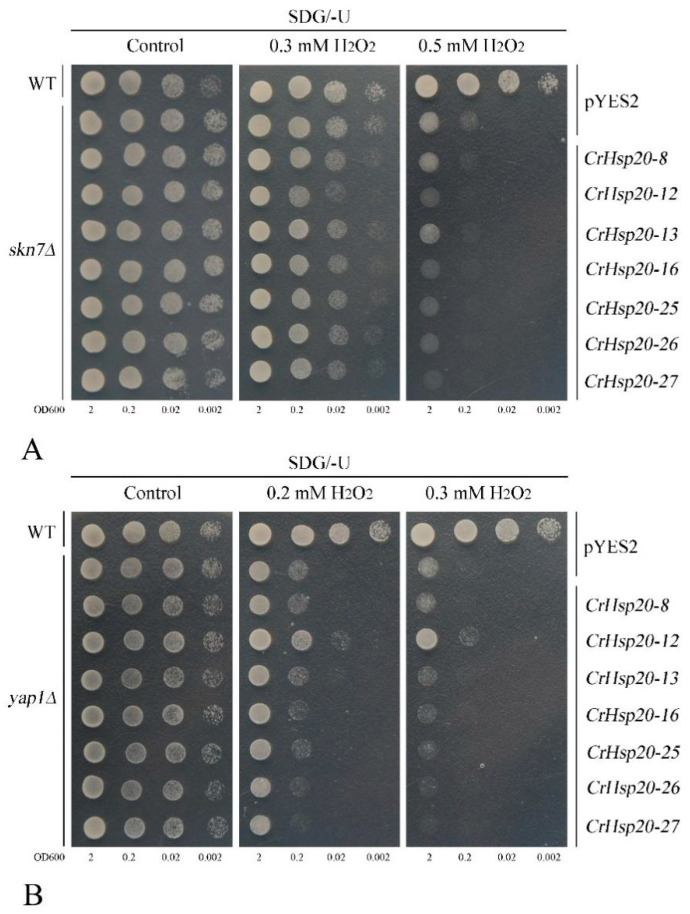
The H_2_O_2_ oxidative stress tolerance confirmations in yeast mutant strains. (**A**) *skn7Δ*. (**B**) *yap1∆*. Yeast cultures were adjusted to OD600 = 2, and 2-μL of serial dilutions (10-fold, from left to right in each panel) were spotted on SDG/−U medium plates supplemented with different concentrations of H_2_O_2_ (0.2 mM, 0.3 mM, and 0.5 mM), and the corresponding yeast spots growing on SDG/−U plates without H_2_O_2_ were used as the control. The plates were incubated for 2–5 days at 30 °C.

**Table 1 ijms-23-06405-t001:** Nomenclature and subcellular localization of heat-shock protein 20 identified from *Canavalia rosea*.

Name	Locus	Protein Length (aa)	*M*_W_ (kD)	pI	II	AI	GRAVY	WoLF_PSORT
CrHsp20-1	01T001428	135	15.67	5.36	63.54	86.67	−0.318	cyto: 7, cysk: 4, extr: 2
CrHsp20-2	01T001854	508	55.89	5.00	48.91	69.09	−0.777	nucl: 14
CrHsp20-3	01T002897	134	14.85	5.39	34.74	82.16	−0.362	chlo: 9, cyto: 2, vacu: 1.5, E.R._vacu: 1.5
CrHsp20-4	01T002898	174	19.98	5.53	44.29	68.16	−0.759	cyto: 14
CrHsp20-5	01T002899	136	15.78	8.96	46.18	62.94	−0.885	cyto: 12, nucl: 1
CrHsp20-6	02T003974	137	15.14	6.09	39.72	105.84	0.331	cyto: 7, chlo: 2, nucl: 2, vacu: 2
CrHsp20-7	02T004687	156	17.69	5.55	42.90	76.86	−0.540	cyto: 9, nucl: 2, cysk: 2
CrHsp20-8	02T004688	156	17.40	5.95	47.07	86.22	−0.480	cyto: 12, chlo: 2
CrHsp20-9	02T004837	152	17.44	6.19	54.71	74.28	−0.666	cyto: 14
CrHsp20-10	02T005143	229	25.57	8.45	38.43	67.64	−0.669	chlo: 13
CrHsp20-11	02T006143	143	16.76	8.76	40.54	90.56	−0.575	cyto: 6, nucl: 3, mito: 2, chlo: 1, E.R.: 1
CrHsp20-12	02T006230	141	16.05	6.13	41.72	74.54	−0.748	cyto: 14
CrHsp20-13	02T007071	159	17.61	6.84	44.93	90.69	−0.371	cyto: 12, chlo: 1
CrHsp20-14	02T007356	485	54.98	9.03	47.46	71.34	−0.702	vacu: 9, golg: 2, nucl: 1, plas: 1
CrHsp20-15	03T007670	212	24.88	5.66	52.16	67.12	−0.824	nucl: 4, cyto: 3, E.R.: 2, golg: 2, mito: 1, vacu: 1
CrHsp20-16	03T010017	159	18.07	7.17	60.79	72.89	−0.596	nucl: 5.5, cyto_nucl: 5, cyto: 3.5, mito: 3, plas: 1
CrHsp20-17	04T013022	193	22.19	5.80	51.69	87.25	−0.490	chlo: 3, cyto: 3, mito: 3, extr: 3, nucl: 1
CrHsp20-18	04T013504	222	25.12	8.73	36.81	84.19	−0.451	cyto: 5, nucl: 4, golg: 2, plas: 1, extr: 1
CrHsp20-19	05T015151	237	26.5	5.60	53.97	75.61	−0.730	chlo: 4, cyto: 4, mito: 4, plas: 1
CrHsp20-20	06T017314	693	78.64	8.63	42.32	73.78	−0.584	cyto: 7, chlo: 2, nucl: 2, mito: 2
CrHsp20-21	06T017504	267	30.47	9.36	38.96	72.62	−0.846	cyto: 4, golg: 3, plas: 2, vacu: 2, chlo: 1, mito: 1
CrHsp20-22	06T017505	285	32.76	7.73	51.98	71.51	−0.809	cyto: 5, chlo: 3, golg: 3, nucl: 2
CrHsp20-23	06T017506	308	35.04	5.11	50.79	64.55	−1.032	vacu: 3, golg: 3, cyto: 2.5, nucl: 2, cyto_pero: 2, chlo: 1, plas: 1
CrHsp20-24	06T017706	139	15.66	5.85	29.32	77.70	−0.670	cyto: 10, extr: 3
CrHsp20-25	06T017707	159	18.02	6.19	63.53	69.18	−0.673	cyto: 13
CrHsp20-26	06T017709	158	17.87	6.18	48.78	70.89	−0.673	cyto: 13
CrHsp20-27	06T018827	144	16.26	8.01	52.70	90.69	−0.435	cyto: 9, chlo: 2, nucl: 2
CrHsp20-28	06T018972	632	71.41	9.43	40.54	97.78	−0.157	plas: 10, nucl: 1, mito: 1, vacu: 1
CrHsp20-29	07T020007	137	15.82	6.43	28.89	92.34	−0.564	cyto: 9, nucl: 2, extr: 2
CrHsp20-30	07T020435	156	17.96	5.56	33.18	65.58	−0.898	mito: 5, cyto: 3, chlo: 2, nucl: 2, plas: 1
CrHsp20-31	07T020816	197	22.5	6.46	33.40	86.50	−0.545	chlo: 5, mito: 5, cyto: 2, vacu: 1
CrHsp20-32	07T020930	147	16.84	9.71	21.75	88.84	−0.256	cyto: 7, nucl: 4, extr: 1, E.R.: 1
CrHsp20-33	08T021644	534	6036	5.57	49.07	54.93	−1.166	chlo: 6, nucl: 4, golg: 2, mito: 1
CrHsp20-34	08T022845	157	17.5	4.71	42.57	83.69	−0.216	cyto: 7, pero: 3, extr: 2, nucl: 1
CrHsp20-35	08T023293	324	36.25	7.12	59.55	80.09	−0.382	chlo: 9, vacu: 3.5, E.R._vacu: 2.5
CrHsp20-36	09T023777	367	41.41	9.29	55.65	63.49	−0.995	nucl: 5, chlo: 4, golg: 3, cyto: 1
CrHsp20-37	09T024521	192	22.31	5.56	43.32	67.45	−0.701	nucl: 9, chlo: 3, plas: 1
CrHsp20-38	11T027728	209	23.78	5.36	66.50	74.98	−0.711	chlo: 5, mito: 5, golg: 2, nucl: 1
CrHsp20-39	11T028183	239	27.27	9.08	51.39	70.17	−0.690	chlo: 12, mito: 2
CrHsp20-40	11T029004	278	32.12	9.48	55.35	59.24	−0.798	nucl: 13
CrHsp20-41	11T029278	219	25	9.20	46.23	79.59	−0.650	cyto: 4, nucl: 3, extr: 2, vacu: 2, golg: 2

*M*_W_: molecular weight; pI: isoelectric point; II: instability index; AI: aliphatic index; GRAVY: grand average of hydropathicity. The *M*_W_, pI, II, AI, and GRAVY values of predicted CrHsp20s were calculated using the ExPASy proteomics server (https://web.expasy.org/protparam/, accessed on 21 March 2022). For the subcellular localization prediction, the online program WoLF_PSORT (https://www.genscript.com/wolf-psort.html, accessed on 21 March 2022) was used.

**Table 2 ijms-23-06405-t002:** Ka/Ks analysis and duplication events for *CrHsp20* genes.

Duplicated Pair	Duplicate Type	Ka	Ks	Ka/Ks	*p*-Value (Fisher)	Positive Selection
*CrsHSP11*/*CrsHSP16*	Segmental	0.432477	1.83043	0.236271	2.34 × 10^−9^	No
*CrsHSP17*/*CrsHSP31*	Segmental	0.151539	0.56805	0.266771	1 × 10^−9^	No
*CrsHSP19*/*CrsHSP38*	Segmental	0.283231	0.711695	0.397967	1.59 × 10^−6^	No
*CrsHSP3*/*CrsHSP4*	Tandem	\	\	\	\	\
*CrsHSP21*/*CrsHSP22*/*CrsHSP23*	Tandem	\	\	\	\	\
*CrsHSP24*/*CrsHSP25*/*CrsHSP26*	Tandem	\	\	\	\	\

## Data Availability

Not applicable.

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
