# Peer review of "Comprehensive Analysis of the Hsp20 Gene Family in Canavalia rosea Indicates Its Roles in the Response to Multiple Abiotic Stresses and Adaptation to Tropical Coral Islands"

_ijms, 2022, doi:10.3390/ijms23126405_

Round 1
Reviewer 1 Report
The Authors analyzed the Heat shock protein 20 (Hsp20) gene family of Canavalia rosea in this manuscript (Manuscript ID: ijms-1745056) using bioinformatics tools, RNA-Seq and quantitative real-time PCR methods under salt, alkaline, osmotic and heat stresses. They found 41 CrHsp20 genes in C. rosea genome. In addition to the predicted protein characterization, the Authors classified CrHsp20 genes in 11 known and 2 unclassified subgroups of Hsp20s using phylogenetic analysis, described their exon-intron structure, conserved motifs, chromosomal localization and cis-acting elements. Furthermore RNA-Seq analyses were performed in different organs of C. rosea plants grown under unstressed condition and after applying abiotic stress treatments. RNA-seq data of 14 CrHsp20s were validated using qRT-PCR and, 7 CrHsp20 cDNAs were cloned in yeast to test their role in heat and oxidative stress. The manuscript describes new information about C. rosea Hsp20 gene family, which can provide basis for further analyses of individual CrHsp20s role in stress responses.
The Authors can find several remarks and questions bellow:
English language is fine, minor spell check is required, and the text need to be corrected in the final manuscript carefully.
The Abstract is too long; Authors should rewrite it and summarize the results emphasizing the novelty of the research.
The Introduction is well written, provides sufficient background information.
Results are presented clearly, but some changes can be made:
In Table 1 GRAVY should be written in one line and the size of letters should be unified in the legend of the table.
Please, unify the abbreviation for isoelectric point; PI or pI.
In 2.2. subsection it was stated, that the “41 CrHsp20s were divided into 12 subfamilies, including 7 cytosol Is (CIs), 5 CIIs, 4 CIIIs, 0 CIV, 2 CVs, 2 CVIs, 1 CVII, 3 mitochondria Is (MIs), 3 MIIs, 1 peroxisome (Po/Px), 4 for the endoplasmic reticulum (ER), and 3 plastids (Ps), based on phylogeny and subcellular localization.” However, none of the CrHsp20s were classified into CIV subfamily. Could the Authors elucidate why did they write 12 subfamilies?
In line 171 AtsHsps is written please, correct it.
Cis-acting regulatory elements were investigated in the upstream region of the CrHsp20 genes. However the length of analyzed sequences is unclear, because in lines 234 and 607 1000 bp is written, but in Figure 4. legend (line 259) 2000 bp is given. Please, correct it.
The whole name of the area, where C. rosea seeds and leaves were collected (South China Botanical Garden and Yongxing Island) should be given not only in Materials and methods section, but when it is first mentioned, too.
According to the Materials and methods samples were collected from the roots, stems, leaves, flowers, and fruits of the plants. Nonetheless, the expression “vine” was used in the Results section and on the Figures 5 and 8. It should be corrected throughout the manuscript, since it is incorrect.
Discussion is acceptable, however references should be added (lines 425-428, 432-435,482-485, 495-500).
Sentence in lines 447-451 is too long please, rewrite it. Also, sentences in lines 459-462 and 500-503 should be checked and reconsidered.
In line 515 instead of “Willow (Salix suchowensis) is a specialized shrub willow…”, “Salix suchowensis is a specialized shrub willow…” could be written.
Detoxification of ROS has been discussed at the end of the Discussion section. Have the Authors investigated the ROS content of the C. rosea plants or in yeast? In my opinion the role of Hsp20 proteins in the ROS elimination should discussed more cautiously and, sentences in lines 557-566 could be reconsidered.
Materials and methods section is acceptable, but it is not clear what the control was in case of stress treatments. It should be added in 4.1. subsection.
In my opinion the manuscript can be accepted after major revision.
Author Response
Comments and Suggestions for Authors from Reviewer 1
The Authors analyzed the Heat shock protein 20 (Hsp20) gene family of Canavalia rosea in this manuscript (Manuscript ID: ijms-1745056) using bioinformatics tools, RNA-Seq and quantitative real-time PCR methods under salt, alkaline, osmotic and heat stresses. They found 41 CrHsp20 genes in C. rosea genome. In addition to the predicted protein characterization, the authors classified CrHsp20 genes in 11 known and 2 unclassified subgroups of Hsp20s using phylogenetic analysis, described their exon-intron structure, conserved motifs, chromosomal localization and cis-acting elements. Furthermore RNA-Seq analyses were performed in different organs of C. rosea plants grown under unstressed condition and after applying abiotic stress treatments. RNA-seq data of 14 CrHsp20s were validated using qRT-PCR and, 7 CrHsp20 cDNAs were cloned in yeast to test their role in heat and oxidative stress. The manuscript describes new information about C. rosea Hsp20 gene family, which can provide basis for further analyses of individual CrHsp20s role in stress responses.
The Authors can find several remarks and questions bellow:
English language is fine, minor spell check is required, and the text need to be corrected in the final manuscript carefully.
Response:
Thank you for your suggestion and we agree with your assessment. We know that there are still a few flaws with the data, methodology, and writing issues, and we have tried our best to improve it and to match the publish criterion. And accordingly, we made some revisions in this manuscript to make it more precise and fluent.
The Abstract is too long; Authors should rewrite it and summarize the results emphasizing the novelty of the research.
Response:
We have streamlined the Abstract part to make it more fluently and compact.
The Introduction is well written, provides sufficient background information.
Results are presented clearly, but some changes can be made:
In Table 1 GRAVY should be written in one line and the size of letters should be unified in the legend of the table.
Response:
We have tried to modify this table to make it more concise and neat.
Please, unify the abbreviation for isoelectric point; PI or pI.
Response:
Sorry for that mistake. They have been uniformly adjusted as "pI".
In 2.2. subsection it was stated, that the “41 CrHsp20s were divided into 12 subfamilies, including 7 cytosol Is (CIs), 5 CIIs, 4 CIIIs, 0 CIV, 2 CVs, 2 CVIs, 1 CVII, 3 mitochondria Is (MIs), 3 MIIs, 1 peroxisome (Po/Px), 4 for the endoplasmic reticulum (ER), and 3 plastids (Ps), based on phylogeny and subcellular localization.” However, none of the CrHsp20s were classified into CIV subfamily. Could the authors elucidate why did they write 12 subfamilies?
Response:
Sorry for that mistake. We deleted “0 CIV”. There are some presentation errors, and we corrected them.
In line 171 AtsHsps is written please, correct it.
Response:
Here this “is” corresponds to the subject "The sequence information for these AtsHsps...”.
Cis-acting regulatory elements were investigated in the upstream region of the CrHsp20 genes. However the length of analyzed sequences is unclear, because in lines 234 and 607 1000 bp is written, but in Figure 4. legend (line 259) 2000 bp is given. Please, correct it.
Response:
Thanks for your reminder. The length of promoter sequences is 1000 bp, and we revised it.
The whole name of the area, where C. rosea seeds and leaves were collected (South China Botanical Garden and Yongxing Island) should be given not only in Materials and methods section, but when it is first mentioned, too.
Response:
Here we added the information for “YX” and “SCBG”.
According to the Materials and methods samples were collected from the roots, stems, leaves, flowers, and fruits of the plants. Nonetheless, the expression “vine” was used in the Results section and on the Figures 5 and 8. It should be corrected throughout the manuscript, since it is incorrect.
Response:
We have adjusted them uniformly as “stem” in the whole manuscript and the figures 5 and 8.
Discussion is acceptable, however references should be added (lines 425-428, 432-435,482-485, 495-500).
Response:
We added some references in these mentioned parts.
Sentence in lines 447-451 is too long please, rewrite it. Also, sentences in lines 459-462 and 500-503 should be checked and reconsidered.
Response:
We revised these sentences.
In line 515 instead of “Willow (Salix suchowensis) is a specialized shrub willow…”, “Salix suchowensis is a specialized shrub willow…” could be written.
Response:
We revised it as your suggestion.
Detoxification of ROS has been discussed at the end of the Discussion section. Have the Authors investigated the ROS content of the C. rosea plants or in yeast? In my opinion the role of Hsp20 proteins in the ROS elimination should discussed more cautiously and, sentences in lines 557-566 could be reconsidered.
Response:
Here we added some explanation as: “On the other hand, although plant Hsp20s have been proved being involved in the ROS pathway [47,48], the Hsp20s participate more on the feedback regulation for ROS scavenging, and the Hsps protection roles cannot alone cope with excessive ROS production under extreme stress’s challenges [48]. That is, the ROS scavenging or ROS toxicity alleviating might be facilitated by accumulating Hsp20 proteins to some extent, while the complete detoxification of ROS has to rely more heavily on the regular-acting cellular antioxidant system, including reductive small molecules (e.g., MTs, LEAs/ASRs) and the antioxidant enzyme system (e.g., superoxide dismutase, peroxidase, catalase).”
Materials and methods section is acceptable, but it is not clear what the control was in case of stress treatments. It should be added in 4.1. subsection.
Response:
We added some description about the C. rosea control plants in 4.1. subsection as: “The C. rosea seedlings placing in 1/2 Hoagland’s solution for different time points (0, 2 h, 48 h) were used as control.”
In my opinion the manuscript can be accepted after major revision.
Response:
Great thanks for your approval! We have made some revisions according to your suggestions, and we do hope this manuscript could meet the criteria of IJMS for publication.

Reviewer 2 Report
The manuscript deals with comprehensive analysis of the Hsp20 gene family in the halophyte plant species in silico. Functional analyses o selected recombinant proteins are also supplied. The reasoning for selection and analysis of given 7 cDNAs should be edited.
What is the reasoning behind using 0.3 mM and 0.5 mM hydrogen peroxide?
The work is elaborated and written well; there are few logical or stylistic errors to be addressed (see detailed below). The work contains some very general claiming, which pretend to be conclusions – these have to be skipped:. e.g. line 336 results.. demonstrate obviously regulated transcriptional patterns; lines 482-484; lines 533-534; 549-550
In Abstract: extremely upregulated – please indicate with more exact value
Line 54 – explain abbreviation (HS)
Line 57 most well – write instead the best
Lines 70-73: text indicated the chaperones are organelles; this is incorrect and should be edited
Line 170 the authors wrote: being found to be acidic in nature; literature/source for this claiming should be added
Line 150 indicate full name for stability (II) for clarity
Line 159 „typical features of nuclear localization“ is misleading: did the authors mean, the two given proteins seem to be localized solely in nucleus?
Line 173 – unclear, what Is mean; pls explain!
Line 180 in apple? Reference is missing or suggests error
Figures: color legends and the figures itself are not very readable (Fig, 2, 3..7, etc). They should be adjusted for better readability and resolution
196-199 the given part is superficial and provides limited information. Extend by some explanation or examples of motifs; in this forms i tis not very informative
Line 253 processed different numbers – unclear, what the authors did mean
Line 276 – other four samples – write instead other tissue types
XY Island – it is some real name or acronym?
For expression data i tis unclear whether absolute values or fold changes are indicated; pls edit in all relevant figures. Legends should explain the sample names/abbreviations as in this form the figures are not self-explanatory
First paraph of Discussion seems to be some remnant from recommendations to authors
The second paraph (lines 419-437) and 467-476 are rather an introduction than discussion!
The authors identified many putatively instable proteins – what is the function/fate of these in tissue during stress?
Methods: 30 days old plants are not really seedlings any more..; uclear reference to replicate: independent experiments, or 3 different plants per variant?
Conclusion: I don’t think the experiment was set to confirm thermotolerance of tested HSPs; I suggest writing: indicate role in thermotolerance
Targets and interplay of HSPs would extend the value of the work; discussion is quite descriptive only.
Author Response
Comments and Suggestions for Authors from Reviewer 2
The manuscript deals with comprehensive analysis of the Hsp20 gene family in the halophyte plant species in silico. Functional analyses o selected recombinant proteins are also supplied. The reasoning for selection and analysis of given 7 cDNAs should be edited.
We have modified some writing errors and made some revisions according to your suggestions. Revised portion have been labeled in the resubmitted manuscript.
What is the reasoning behind using 0.3 mM and 0.5 mM hydrogen peroxide?
Response:
According to some previous reports, such as Zhou et al (https://doi.org/10.1186/s12870-022-03587-9), Jagadish et al (https://onlinelibrary.wiley.com/doi/10.1111/pce.14050), and Jiang et al (https://onlinelibrary.wiley.com/doi/epdf/10.1111/j.1365-3040.2009.01987.x), plant Hsp20s might be involved in stress resistance by mediating ROS-scavenging pathway or against ROS damage. In wild type Pichia pastoris yeast, the constitutive expression of RcHSP17.8 could significantly improve the tolerance of yeast to 6 mM H2O2 (Jiang et al, 2009). Here we firstly tried the oxidative stress tolerance of CrHsp20s with two mutant strains, skn7∆ and yap1∆, which are sensitive to 0.5 mM and 0.3 mM H2O2. Although the results showed negative tolerance of several CrHsp20s to H2O2 in yeast, this research also indicated that the molecular chaperones like Hsp20s might act in different functional ways from other chaperones, such as MTs (https://linkinghub.elsevier.com/retrieve/pii/S017616172100198X) or LEAs/ASRs (https://www.mdpi.com/1422-0067/22/9/4554), in mediating abiotic stress resistance.
The work is elaborated and written well; there are few logical or stylistic errors to be addressed (see detailed below). The work contains some very general claiming, which pretend to be conclusions-these have to be skipped:. e.g. line 336 results.. demonstrate obviously regulated transcriptional patterns; lines 482-484; lines 533-534; 549-550
Response:
We added the description for our results in the detail in our revised manuscript.
In Abstract: extremely upregulated-please indicate with more exact value.
Response:
We revised it as: “...most CrHsp20 genes were greatly upregulated (up to dozens to thousands of times).”
Line 54 – explain abbreviation (HS).
Response:
“HS” means “Heat stress”, it has been marked at the beginning of the introduction part.
Line 57 most well-write instead the best.
Response:
We revised it. Thanks.
Lines 70-73: text indicated the chaperones are organelles; this is incorrect and should be edited
Response:
Thanks, here “act in...” might be a bit of a misnomer, and it means “work in...”.
Line 150 the authors wrote: being found to be acidic in nature; literature/source for this claiming should be added.
Response:
Here it means that most of CrHsp20s are acidic in nature due to their pI values are less than 7, according to the protein characteristics listed in Table 1. We added “Table 1” information here.
Line 150 indicate full name for stability (II) for clarity.
Response:
The II means instability index, which was indicated at the first lines of this paragraph.
Line 159 “typical features of nuclear localization“ is misleading: did the authors mean, the two given proteins seem to be localized solely in nucleus?
Response:
We revised it according to your advice.
Line 173 – unclear, what Is mean; pls explain!
Response:
We revised it as: “these 41 CrHsp20s were divided into 11 subfamilies, including 7 cytosol Is (CIs), 5 CIIs, 4 CIIIs, 2 CVs, 2 CVIs, 1 CVII, 3 mitochondria Is (MIs), 3 MIIs, 1 peroxisome (Po/Px), 4 for the endoplasmic reticulum (ER), and 3 plastids (Ps), based on phylogeny and subcellular localization.”
Line 180 in apple? Reference is missing or suggests error.
Response:
Here it refers to C. rosea, and we revised it.
Figures: color legends and the figures itself are not very readable (Fig, 2, 3..7, etc). They should be adjusted for better readability and resolution.
Response:
We uploaded the figures separately after the manuscript submission, and we hope the editors could replace the poor figures with better figures.
196-199 the given part is superficial and provides limited information. Extend by some explanation or examples of motifs; in this forms it’s not very informative.
Response:
We added: “In addition to the genes’ structures, the conserved motifs in protein sequences of family members also indicated the conservatism and variability of this gene family. ”
Line 253 processed different numbers-unclear, what the authors did mean.
Response:
We adjusted “ different numbers of...” as “a number of ...”.
Line 276 – other four samples – write instead other tissue types.
Response:
Here we added: “including root, stem, flower, and young fruit tissues”.
XY Island – it is some real name or acronym?
Response:
Here we added: “(a coral island located in South China Sea, a native habitat for C. rosea plants)”.
For expression data it is unclear whether absolute values or fold changes are indicated; pls edit in all relevant figures. Legends should explain the sample names/abbreviations as in this form the figures are not self-explanatory
Response:
We added some information in these figure legends according to your suggestions.
First paraph of Discussion seems to be some remnant from recommendations to authors
Response:
Sorry about that, and we deleted it.
The second paraph (lines 419-437) and 467-476 are rather an introduction than discussion!
Response:
Here we just want to emphasize the significance of this study, since C. rosea is just a unfamiliar wild plant species without famous economic value, but it is indeed a pioneer species for coral reef and island greening in tropical region. The heat resistant of C. rosea is typical, while the related molecular mechanisms are neither clear nor certain.
The authors identified many putatively instable proteins-what is the function/fate of these in tissue during stress?
Response:
In general, the instability is the basic feature of proteins being stress responsive molecules, since any stress is rapidly changed along with the environment surrounding individuals. When the environmental conditions become normal, these instable proteins tend to be degraded.
Methods: 30 days old plants are not really seedlings any more..; uclear reference to replicate: independent experiments, or 3 different plants per variant?
Response:
The seed of C. rosea germinated very slowly, and the growth of C. rosea seedlings is slow too in greenhouse. The adult C. rosea plants are perennial, and 30 days old plants are seedlings.
Conclusion: I don’t think the experiment was set to confirm thermotolerance of tested HSPs; I suggest writing: indicate role in thermotolerance
Response:
We modified the expression according to your suggestion.
Targets and interplay of HSPs would extend the value of the work; discussion is quite descriptive only.
Response:
We made some supplemental information and related references in the discussion part, we do hope to meet with approval.

Round 2
Reviewer 1 Report
During the revision of the manuscript (Manuscript ID: ijms-1745056) the Authors addressed my earlier comments and the manuscript have been corrected, according to my suggestions. In my opinion, the manuscript can be accepted in its present form.